# The Possible Role of the Microbiota-Gut-Brain-Axis in Autism Spectrum Disorder

**DOI:** 10.3390/ijms20092115

**Published:** 2019-04-29

**Authors:** Piranavie Srikantha, M. Hasan Mohajeri

**Affiliations:** Department of medicine, University of Zurich, Winterthurerstrasse 190, 8057 Zürich, Switzerland; piranavie.srikantha@uzh.ch

**Keywords:** Autism, ASD, microbiota, gut-brain-axis, microbiota-gut-brain-axis, probiotics, prebiotics

## Abstract

New research points to a possible link between autism spectrum disorder (ASD) and the gut microbiota as many autistic children have co-occurring gastrointestinal problems. This review focuses on specific alterations of gut microbiota mostly observed in autistic patients. Particularly, the mechanisms through which such alterations may trigger the production of the bacterial metabolites, or leaky gut in autistic people are described. Various altered metabolite levels were observed in the blood and urine of autistic children, many of which were of bacterial origin such as short chain fatty acids (SCFAs), indoles and lipopolysaccharides (LPS). A less integrative gut-blood-barrier is abundant in autistic individuals. This explains the leakage of bacterial metabolites into the patients, triggering new body responses or an altered metabolism. Some other co-occurring symptoms such as mitochondrial dysfunction, oxidative stress in cells, altered tight junctions in the blood-brain barrier and structural changes in the cortex, hippocampus, amygdala and cerebellum were also detected. Moreover, this paper suggests that ASD is associated with an unbalanced gut microbiota (dysbiosis). Although the cause-effect relationship between ASD and gut microbiota is not yet well established, the consumption of specific probiotics may represent a side-effect free tool to re-establish gut homeostasis and promote gut health. The diagnostic and therapeutic value of bacterial-derived compounds as new possible biomarkers, associated with perturbation in the phenylalanine metabolism, as well as potential therapeutic strategies will be discussed.

## 1. Introduction

The microbiota-gut-brain-axis describes the bidirectional physiological connection to exchange information between the microbiota, the gut and the brain [1]. The gastrointestinal (GI) tract, the largest surface in the body, meets trillions of microorganisms separated by the gut barrier [2]. The gut barrier is made out of the commensal gut microbiota, a mucus layer and epithelial cells connected through tight junctions [3,4]. A healthy microbial composition is important to health, as dysbiosis is often observed in gut-related diseases and conditions such as inflammatory bowel disease, irritable bowel syndrome, diabetes and obesity, but also in unconventional diseases such as acne, allergy, cardiovascular disease, stress, depression, Alzheimer’s disease, multiple sclerosis, Parkinson’s and autism spectrum disorder (ASD) [5,6,7,8,9,10,11,12]. Dysbiosis is an altered microbial composition favouring pathogenic microbes over beneficial ones in the gut. Knowing that dysbiosis at a young age affects health-status during later life, it is important to study the life-long impacts of compromised GI functions during childhood. 

In addition to dysbiosis, GI symptoms are four times more prevalent in children with ASD compared to the normal population [13]. A wide range of GI symptoms are found in these children, such as constipation, diarrhoea, bloating, abdominal pain, reflux, vomiting, gaseousness, foul smelling stools and food allergies [1,14,15,16,17,18,19]. In addition, lower bacterial diversity found in autistic children [20,21] could be related to the severity of GI symptoms [20]. The higher abundance of *Clostridia* species (spp) in autistic individuals suggests involvement in the pathogenesis of ASD [22]. Structural changes in the brain [23,24,25,26,27,28,29,30] and less sociable behaviours [31,32,33,34] observed in germ-free mice indicate a functional connection between the microbiota and the brain. However, as there are major differences between the murine and the human microbiota [35], the validity of drawing conclusions from GF mice to humans is limited. Nevertheless, the observed behavioural changes in germ-free mice raise the question regarding the mechanisms of action of microbiota on human metabolism.

The frequently reported comorbidity of GI problems in autistic children [1,14,15,16,17,18,19] motivated the central question of this review: is there a connection between the gut microbiota and the central nervous system (CNS) affecting the pathology of ASD? This is an important question considering that only approximately one third of all autistic cases could be linked to genetic causes [36]. Furthermore, no satisfactory effective treatment is available for Autism Spectrum Disorder to date. Therefore, understanding and modulating the microbiota-gut-brain-axis may be an effective and promising way for ameliorating the disease symptoms [37]. The objective of this review is to summarize all peer-reviewed human studies and reviews on the topic connecting gut microbiota with ASD and suggesting metabolites, which could serve as potential biomarkers after further validation. The elevation of 3-(3-hydroxyphenyl)-3-hydroxypropionic acid, 3-hydroxyphenylacetic acid and 3-hydroxyhippuric acid in autistic children hinted to perturbation in the phenylalanine metabolism [38]. These metabolites are related to the abundance of *Clostridia* spp which worsens autistic behaviour [38].

## 2. Material and Methods

In this systematic review we concentrated on the question: Is there a connection between the gut microbiota and the central nervous system (CNS) affecting the pathology of ASD? PubMed databank and online books were searched on February 1st 2018. The objective was to study the possible connection between the microbiota/microbiome and ASD. The following search parameters and MeSH terms “Microbiome”, “microbiota”, “gut-brain-axis”, “autism”, “autistic disorder” and “human” delivered 139 hits (Figure 1). 

We concentrated on bacterial taxa excluding excluded citations regarding viruses and archaea. Thus, inclusion criteria were the following: -Diagnosis of ASD by medical experts-Human studies of the microbiome or metabolome in autistic individuals-ASD individuals were compared to a matched control group-Collection and analysis of at least one of the following biomaterials: GI biopsies, faecal, urinary or blood samples-Analysis of bacterial genome by sequencing-Published in a peer-reviewed article-Availability of the full text publication-Availability of the paper in English

Most of the examined papers were published between 2013 and 2018. The evaluation of literature was conducted by two independent researchers resulting in the same included papers. Seven papers were excluded as full texts were not available. Five non-English papers were excluded, and five papers were excluded due to the lack of any relevance to the topic. In total, 136 papers were examined for this systematic review. Online books were used for the definitions of diseases.

## 3. Comorbidities in Autism Spectrum Disorder

### 3.1. Definition

Autism spectrum disorder (ASD) is a collective term for autism, Asperger’s syndrome and pervasive developmental disorder not otherwise specified [40]. More male than female individuals are affected by the disease [41,42]. The prevalence of ASD was one in fifty-nine children in the United States in 2014 [43,44]. Autistic behaviour is characterized by poor communication skills, social withdrawal, repetitive or restrictive pattern in behaviour, interests and activities and the onset of these diagnostic features in early developmental stage [45]. Other reported behaviour are picky eating habits [16,24,46], increased aggression [47] and anxiety [48]. Regressive autism or late onset autism describes a subgroup of patients with initially normal development but with a gradual loss of acquired skills in communication or social interaction [40]. Asperger’s Syndrome manifests next to social withdrawal with superior cognitive skills. The phenotypes of ASD are heterogeneous, suggesting that many factors play into the aetiology [27]. Both genetic and environmental factors are implicated in ASD but most cases of ASD are idiopathic [20]. Lastly, abnormal eating habits are frequently reported in autistic individuals, possibly leading to deficiencies in vitamins, minerals and fatty acids [16].

### 3.2. Gastrointestinal Symptoms

Several comorbidities are frequently reported in ASD children (Figure 2). GI problems are often observed in autistic individuals [1,14,15,16,17,18,19]. In a small sample size including ten autistic children, nine non-autistic siblings and ten healthy children severity and number of GI symptoms were found to be significantly higher in autistic children and their siblings compared to healthy controls [49]. Accordingly, a different bigger study comparing 230 pre-schoolers discovered similar results. ASD patients were suffering significantly more often from GI problems than healthy controls. ASD patients with GI symptoms were found to have more anxiety problems and other somatic complaints in addition to less social interaction in comparison to ASD patients without GI problems [14]. Constipation was found to be the most common GI symptom observed in autistic children. The same authors also found a connection between rigid-compulsive behaviour and the occurrence of constipation [13]. Additionally, GI problems in autistic children were reported to lead to more tantrums, aggressive behaviour and sleep disturbances, further worsening the behaviour compared to autistic individuals without GI symptoms [15]. One review suggested that altered behaviour such as aggression, self-injury or sleep disorder observed in autistic children might be an expression of abdominal discomfort [50].

Autistic children show an altered metabolism and absorption of disaccharides in their gut epithelium (Figure 2) [20,51]. The disaccharidases in the brush border of the ileum such as lactase, maltase glucoamylase and sucrose isomaltase, were found to have lower mRNA levels and hence a decreased gene expression [52,53]. The sodium-dependent glucose cotransporter (SGLT1) and the glucose transporter 2 (GLUT2) transport glucose, galactose and fructose actively across the luminal and basolateral membranes of the enterocytes. Autistic children were reported to exhibit significantly decreased mRNA levels of both hexose transporters in the ileum [53]. Consequently, there is a malabsorption in the small intestine and more mono- and disaccharides enter the large intestine. Thus, bacteria fermenting these low-molecular sugars profit and outcompete bacteria degrading polysaccharides leading to an altered microbial composition in the GI tract. Higher amounts of sugars in the large intestine could lead to osmotic diarrhoea or these sugars could serve as substrates for the production of gases [53]. Diarrhoea and bloating are both GI symptoms observed in ASD patients. GI symptoms in ASD patients are known to correlate with the severity of autistic behaviour [15].

### 3.3. Increased Gut Permeability

An increased intestinal permeability was observed in ASD patients (Figure 2) by measuring lactulose in blood after oral administration [15]. Autistic children as well as their non-autistic siblings were found to have increased gut permeability compared to controls [54]. Increased gut permeability is probably a result of decreased expression of barrier forming proteins and the elevated expression of pore forming proteins of tight junctions. These altered expressions were found by analysing mucosal samples from the small intestine of autistic children compared to controls [55]. Changed permeability could lead back to lower counts of *Lactobacillus* strains in autistic patients, as they are associated with the maintenance of tight junction in the epithelial barrier of the intestines [15]. Bacterial metabolites, such as LPS, can easily pass the intestinal barrier and cause an inflammation affecting the brain through altering cytokine levels [1]. Increased intestinal permeability was also found in some close non-autistic relatives of autistic individuals, suggesting that intestinal integrity is not a consequence of ASD. Rather, increased gut permeability should be viewed as a cause, which could in combination with other environmental factors, at least partially, lead to the pathology of ASD [56].

### 3.4. Alterations in the Brain

The analysis of post-mortem brains of autistic patients revealed altered expression of some claudines (CLDN) in the blood-brain-barrier (BBB) of the cortex and cerebellum. Significant changes were observed in the expression of the genes of CLDN-5 and CLDN-12. These two major constituents of tight junctions were overexpressed in the cortex and cerebellum of autistic children compared to controls (Figure 2). CLDN-5 is an important protein in tight junctions for cell adhesion of endothelial cells in the brain. However, the protein expression of CLDN-12 was significantly reduced in the cortex of autistic children even though the gene was overexpressed [55]. The over-expressions of the genes are suggestive of either a higher integrity of the BBB in autistic children or a compensatory mechanism to uphold the integrity of the BBB. The second explanation appears more plausible, as an increased destruction of the protein CLDN-12 could lead to an over-expression of the CLDN-12 gene and other tight junction components such as CLDN-5. These results suggest a lower integrity of the BBB leading to a compensatory overexpression of tight junction components. 

Increased activation of microglial cells and other structural changes have also been observed post-mortem in the brains of autistic individuals (Figure 2) [27,57]. The increase in activation of microglial cells suggests that ASD is a pathology caused or at least accompanied by immune activation in the brain. Inflammation in the brain could subsequently lead to malfunctioning synapses [27]. During inflammation the brain releases arginine vasopressin, a metabolite, which is known to act on social behaviour and is a considered biomarker for ASD [27]. A structural change was the increased weight of ASD brains possibly due to the presence of a larger number of neurons in the prefrontal cortex [57]. Moreover, Purkinje cells are found in decreased numbers in the cerebellum of ASD patients [43,58]. Purkinje cells produce GABA, which makes them vulnerable to the tetanus neurotoxin produced by *Clostridia tetani* [43]. Elevated amounts of *Clostridia* spp is reported in ASD, thus could account for decreased Purkinje cells in the cerebellum of ASD individuals.

There is a bidirectional relationship between the gut and the brain including among others vagal fibers [59]. Enteroendocrine cells in epithelial barrier of the gut can sense the composition of the lumen such as nutrients and bacterial metabolites. These cells synapse with afferent vagal fibres, thus directly connecting the intestinal milieu with the brainstem [60]. Peripheral stimulation of vagal fibres leads to a dopamine release in the reward system, as was shown in vivo [61].

### 3.5. Mitochondrial Dysfunction

Mitochondrial dysfunction was detected in ten ASD children (Figure 2) comparing intestinal mucosa samples of rectum und caecum with that of ten Crohn’s disease children (representing non-autistic children with GI symptoms) and ten healthy children [62]. ASD children displayed lower activity of the electron transport chain (ETC) enzyme complex IV and citrate synthase in mucosal samples taken both from rectum and caecum. In addition, they also showed significantly higher quantity of ETC complex I in the mucosa of the caecum. The quantities of the ETC enzyme belonging to complexes III, IV and V were only significantly increased in the caecum [62]. The occurrence of these changes only in the caecum and not also in rectum is intriguing. Especially so, since the rectum acts as a transition passage, whereas the caecum inhabits many bacteria species. The authors discuss their finding stating that the altered expression of these enzymes, all involved in the energy production, leads to mitochondrial dysfunction. Mitochondrial dysfunctions disrupt normal functioning of enterocytes and therefore cause gut-dysmotility and higher sensitivity to oxidative stress. Increased oxidative stress results in damaged proteins and lipids in the cell and consequently causes diminished function of enterocytes. This suggests that bacterial metabolites may affect mitochondrial function in the caecum. SCFAs produced by *Clostridia* species spp could enter the mitochondria and be utilized as substrates for energy production [62]. For example, butyrate is converted into acetyl-CoA, which then is utilized in the citric cycle for NADH production. NADH, on the other hand, is a substrate for ETC complex I, which could be the factor elevating its activity. Additionally, gut-dysmotility caused by mitochondrial dysfunction would explain constipation observed in autistic individuals [62]. The occurrence of many GI problems observed in autistic children could partially be explained by gut-dysmotility. Interestingly, GI problems similar to that of autistic children have also been noted in children with mitochondrial dysfunction [62], again suggesting a link between mitochondrial dysfunction, GI problems and microbiota in ASD individuals. 

## 4. Factors Influencing the Gut Microbiota

### 4.1. Definitions and Facts

The term microbiota describes the entirety of all bacterial, archaeal and fungal microorganisms living on the skin, in the mouth, in the respiratory, GI and vaginal tracts [37]. As the term microbiota describes all microorganisms living in and on humans, the term microbiome describes all the genes expressed in all these microorganisms [50]. The highest diversity of microbiota is found in the large intestine [17]. Approximately 1000 different spp of bacteria are reported to live in the GI tract [63]. Healthy gut microbiota interacts with the human metabolism by educating the immune system, protecting against pathogenic microbes, metabolising non-digestible carbohydrates, producing essential vitamins and antimicrobial substances, stimulating angiogenesis and importantly reinforcing the gut barrier through increased mucine production [15,40,64,65,66,67]. The microbial composition varies through age, diet, diseases, geography [37,68] and shared environment [34]. 

### 4.2. Developmemt and Disruptions of Microbial Colonisation in Autism

Colonisation of the GI tract with microbiota starts prenatally as microorganisms have been detected in the placenta and meconium [17,28,69,70]. Moreover, some bacterial species of *Enterococcus*, *Streptococcus* and *Staphylococcus* were found in the blood of the umbilical cord. This suggests a transfer of microbiota from the mother to the foetus [71]. A new-born shows an instable and highly dynamic intestinal microbiota with less microbial diversity and a dominance of the phyla Proteobacteria in the gut. With time, the diversity and colonisation increase and the dominance in Firmicutes and Bacteroidetes can be observed. [72]. First colonisers of healthy new-borns are *Enterobacteria*, *Staphylococcus* and *Streptococcus* spp, which are facultative anaerobes. These bacteria consume the oxygen in the gut and create an anaerobic environment. Following these first colonisations strict anaerobes as *Bacteroides*, *Bifidobacterium* spp and *Clostridium* spp can colonise the gut [72]. The composition of the microbiota stabilises around the age of 2–3 years [17,73]. Interestingly the brain of neonates also grows from 36% to approximately 90% of its future adult volume until the age of two. Also, the formation of new synapses in the brain peaks between the age of three months and two years [26]. Thus, the critical window for establishment of a healthy microbial composition falls into the same critical time window for brain development [26]. The developing brain is sensitive to many external and internal environmental factors. Therefore, it is postulated that changes in prenatal maternal stress level, infections or diet can play a role in the pathophysiology of neurodevelopmental disorders [71,74]. 

Preterm birth, mode of delivery and breastfeeding influence the gut microbial composition of neonates. Vaginally born babies who are breastfed have the healthiest composition of microbiota. Beneficial bacteria such as *Bifidobacteria* are more abundant in these babies than pathogenic bacteria including *Clostridium difficile* or *Escherichia coli* [75]. Human breast milk promotes the growth of *Bifidobacterium longum* [69]. Vaginally born babies have a microbiota that resembles vaginal microbiota composition of the mother with dominance of *Lactobacillus*, *Prevotella* and *Snethia* spp [76]. Neonates born via caesarean section have an altered microbiota composition resembling the microbiota of the mother’s skin with dominance of *Staphylococcus*, *Corynebacterium*, *Propionibacterium* spp [76,77], *Escherichia coli* and *Clostridium difficile* [78]. Exclusively breastfed infants display also richness in *Bifidobacterium* spp, which are specialised in digesting oligosaccharides contained in human milk [76]. A study found that autistic children experienced significantly shorter period of breastfeeding [79]. Preterm born babies possess low levels of gastric acidity, as GI functions have not fully matured. Consequently, more pathogenic microbes survive the stomach and can colonise the intestines leading to an accumulation of pathogenic microbiota and diminished degree of microbiome diversity in the GI tract [6]. They display dominance in Proteobacteria and lack in the genera *Lactobacillus* and *Bifidobacterium* [73].

Using information from a population-based registry with approximately 2.6 million children, a Swedish study noted that elective caesarean births in comparison to natural vaginal delivery is associated with a significantly 20% higher risk of the child developing ASD [80]. In contrast, no association was found in the sibling control analysis, so the authors concluded that there was no link between caesarean birth and ASD at all [80]. Similarly, another cohort study from the UK with over 18,000 participants also found no association found between caesarean birth and development of ASD [81]. 

Antibiotic treatment during the first three years of life can have detrimental effect on normal microbiota establishment with long-term effects [82]. It is known that many autistic children take relatively high doses of oral antibiotics in early years of life, which could be a factor disturbing their gut microbiota [20]. Recovery of the microbiota to the pre-treatment state is incomplete even several months after the cessation of the treatment [37]. Early treatment of infants with antibiotics can therefore disrupt the colonisation of gut with beneficial bacteria [73]. Changes in the composition of the microbiota could lead to a leaky gut or increased passage of metabolites such as LPS or SCFAs through intestinal barrier [1,83], leading to the activation of an inflammatory response through binding to Toll-like receptor 4 also affecting the CNS [1].

Maternal infection during pregnancy can lead to a modified microbial composition as well as an increased risk of ASD. Women were found to have a higher risk of carrying out autistic children, when they were hospitalised due to an infection during pregnancy. Interestingly, the risk for ASD was higher when infection occurred during a critical time window of development. Viral infections in the first trimester of the pregnancy and bacterial infections in the second trimester were associated with a higher risk of birthing autistic children [27]. These data are in agreement with results obtained in maternal immune activated (MIA) mice. Pregnant mice were injected with polyinosinic:polycytidylic acid (poly(I:C)) to simulate viral infections. The offspring displayed altered microbial composition, GI barrier defects and behaviour similar to autistic features [5]. Elevated levels of cytokines in maternal blood were proposed to be the causing factor as poly(I:C) is also a cytokine [5]. Elevated levels of IL-6 were found in the colon from MIA mice [5]. The cytokine IL-6 is known to alter the expression of tight junction proteins, as it is proinflammatory, thus helping white blood cells to migrate through epithelial barriers [5]. Hence, the increased gut permeability in autistic children could be a consequence of immune activation in the GI tract [84,85]. Bacteroides, Bifidobacterium spp and Lactobacillus spp create an anti-inflammatory milieu, whereas Clostridium spp stimulate a pro-inflammatory milieu [86]. Cytokines produced by microbiota affected the oligodendrocytes functioning [23]. Therefore, an inflammation during pregnancy can result in alteration of myelination in the brain and possibly in behavioural changes, as observed in ASD [23]. 

Taken together, a higher risk for developing ASD was observed in children (i) who were born pre-term with a very low birth-weight [69], (ii) who were born by caesarean delivery, (iii) who were not breastfed, (iv) who underwent prolonged hospitalization, (v) who were treated with antibiotics for a long period [16] or (vi) whose mothers had an infection during pregnancy [27]. 

## 5. Alterations in Microbial Composition and Metabolic Profile in Autistic Individuals

### 5.1. Changes in Microbial Composition in Autistic Children

A regularly observed phenomenon in the faeces of autistic children is a significantly decreased ratio between the phyla Bacteroidetes to Firmicutes [49,53,76,87], which pointed to elevated numbers of Firmicutes in contrast to decreased levels of Bacteroidetes. Conflicting data were reported by De Angelis et al., who found in a study with thirty subjects that Firmicutes counts were lower than Bacteroidetes in autistic children when compared to healthy controls [40]. These results, however, were not statistically significant [40]. Moreover, the phyla Fusobacteria and Verrucomicrobia were also represented in lower concentrations (Figure 3) in the faeces of ten autistic children compared to ten healthy controls [40]. Significantly elevated bacteria in autistic children were *Akkermansia muciniphila*, *Anaerofilum*, *Barnesiella intestinihominis*, *Clostridium* spp, *Dorea* spp, the family Enterobacteriaceae, *Faecalibacterium* spp (especially *Faecalibacterium prausnitzii*), *Roseburia* spp, *Parasutterella excrementihominis*, *Prevotella copri*, *Prevotella oris* and *Turicibacter* spp. *Escherichia coli* was significantly decreased in autistic children [40]. Other higher represented spp were *Aeromonas*, *Odirobacter splanchnicus*, *Parabacteroides*, *Porphyromonas*, *Pseudomonas*, and *Turicibacter sanguinis*. Significantly decreased bacteria in autistic children were *Bifidobacterium*, *Fusobacterium*, *Oscillospira*, *Sporobacter*, *Streptococcus* and *Subdoligranulum*. Lower represented genera in autistic children were *Collinsella* spp except *Collinsella aerofaciens*, *Enterococcus* spp, *Lactobacillus*, *Lactococcus* and *Staphylococcus* [40]. Kang et al. compared the intestinal flora of twenty autistic children with GI problems to the intestinal flora of twenty neurotypical children. They found that the significant lower bacterial diversity found in autistic children [20,21] correlates with the severity of GI symptoms. No major differences were detected at phylum levels; however, this discrepancy could be a result of the parents collecting and freezing the stool samples and not the research team. Non-autistic children had a significantly higher abundance of the genus *Coprococcus* and class Prevotellaceae compared to autistic faecal samples. *Prevotella* spp are commensal gut microbes, specialised in degrading plant polysaccharides and synthesising vitamin B1. Lower abundance of *Prevotella* spp could therefore result in a vitamin B1 deficiency. In addition, Veillonellaceae were found in lower abundance in autistic children. Small differences were observed in *Sutterella* genus, being less abundant in autistic children [20]. These data reveal a change in microbiota composition in autistic children that may have biochemical and functional consequences on the host. 

Intestinal biopsies from the caecum and the terminal ileum of twenty-three ASD children showed significantly increased numbers of *Sutterella* spp compared to nine healthy controls. *Sutterella* spp are normally scarce in a healthy microbiota and were found in none of the controls, but in twelve of the twenty-three ASD patients [88]. The microbial composition was also studied in duodenal biopsies comparing nineteen autistic children with twenty-one healthy controls [89]. The study did not find any major differences between the phyla, but some alterations were observed at genus und species levels. The genus *Burkholderia* was significantly increased in autistic participants compared to controls. The genera *Actinomyces*, *Oscillospira*, *Peptostreptococcus* and *Ralstonia* were elevated and the genus *Neisseria* was significantly under-represented in autistic children compared to controls. The genera *Bacteroides*, *Devosia*, *Prevotella* and *Streptococcus* were also decreased. Also, lower abundances of *Escherichia coli* were observed [89]. Kushak et al. could not find major differences at phylum level as above-mentioned studies [40,49,53,76,87]. This could be due to the fact, that all the other studies investigated microbial composition from stool samples and not biopsies from duodenal mucosa. The stool samples represent, by and large, microbial composition of the large intestine [90]. Kushak et al., on the other hand, represented microbial composition of the small intestine. Kushak’s results suggested that microbial changes in the large intestine could have a greater impact on the pathology of ASD than the changes in the small intestine. Therefore, a better representation of the microbial gut milieu could be achieved by collecting mucosal samples from all over the GI tract for comparison of the microbiota resulting in a better understanding of the inter-relationships of the microbiota functioning and the interaction with the host. 

Biopsies from the ileum and the caecum of fifteen ASD children (seven controls) showed a lower abundance of Bacteroides leading to a significantly higher Firmicutes to Bacteroidetes ratio [53]. Levels of Clostridiales (Firmicutes) were slightly elevated in ASD group, especially the families Lachnospiraceae and Ruminococcaceae and the genus *Faecalibacterium*. The class Betaproteobacteria (Proteobacteria) was significantly higher abundant in faecal samples of the caecum in ASD patients, within the class Betaproteobacteria the family Alcaligenaceae was the highest abundant [53]. Concomitant with increasing levels of Bacteroidetes the mRNA levels of SGLT1 in ileum and caecum significantly increased, whereas the mRNA levels of sucrase isomaltase (disaccharidase) decreased in the caecum. With increasing mRNA levels of sucrose isomaltase a significant decreased amount of Firmicutes was observed in the caecum [53]. These associations may be of therapeutic interest, as they demonstrated how bacterial composition influenced expressions of the host’s sugar transporters and enzymes to consequently alter their nutrients’ availability. Therefore, directing the bioactive therapeutic molecules to the small intestine or the colon may be a practical approach to circumvent systematic absorption achieving a specific modulation of the microbiota in these organs.

Sibling control studies delivered discordant results. A Slovakian study enrolling twenty-nine participants including autistic children, their siblings and healthy children found that the amount of *Lactobacillus* spp was significantly increased (Figure 3) in the faeces of autistic children compared to siblings and healthy controls. Amount of *Desulfovibrio* and *Clostridia* spp were also increased, whereas *Bifidobacterium* spp were decreased in autistic children (Figure 3) [49]. The authors associated the severity of behavioural symptoms in social interaction, communication and restricted/repetitive behaviour in ASD with the amount of *Desulfovibrio* spp present. Therefore, *Desulfovibrio* spp were considered pathogenic microbes. Interestingly, non-autistic siblings of autistic children showed lower abundance of Bacteroidetes and higher abundance of Firmicutes compared to healthy controls with no family history of ASD [49]. Furthermore, non-autistic siblings had lower abundance of *Clostridia* and *Desulfovibrio* spp and significantly lower counts of *Bifidobacterium* spp compared to autistic children. Therefore, it was postulated that the levels of bacterial spp could be the tipping factor between autistic or healthy phenotypes [49]. In contrast, Gondalia et al. found no significant difference in microbial composition after analysing fifty-one autistic children and their fifty-three non-autistic siblings [91]. Similarly, a different study including fifty-nine autistic children and forty-four non-autistic siblings also found no significant difference in the composition of the gut microbiota [92]. These data suggest that the microbiota of siblings are similar and independent from their autistic phenotype [91], probably as a result of the shared environment and genetic makeup [93]. 

Higher occurrence of *Clostridium* spp in the gut was associated with disease severity using the Childhood Autism Rating score (CARs score) [15]. Significantly higher counts of the bacterium *Clostridium perfringens* were found in faecal samples of thirty-three autistic children compared to thirteen controls. Especially the spp *Clostridium perfringens* producing the beta2-toxin gene was significantly elevated in autistic children. The authors also found a significant relationship between the abundance of the beta2-toxin gene and occurrence of ASD [21]. Herbicides were considered to preserve *Clostridia* spp and harm beneficial bacteria [94]. A review suggested that a sub-acute tetanus infection with a *Clostridium* spp might be the cause of some cases of ASD. Infection with the pathogen *Clostridium tetani* only occurs in dysbiotic GI tracts, as it is an opportunistic pathogen. It may be inactive for several months until favourable conditions allow its growth. *Clostridium tetani* produces tetanus neurotoxin, which crosses the intestinal barrier. Tetanus neurotoxin is then transported via the vagal nerve to the nucleus solitarius and subsequently to the whole CNS. Tetanus neurotoxin inhibits the release of synaptic vesicles containing neurotransmitters by irreversibly cleaving synaptobrevin, a membrane-associated protein, which is important for vesicle stability. Synapses with cleaved synaptobrevin degenerate and lower synaptic activity correlates with diminished social behaviour found in ASD (Figure 4). Tetanus neurotoxin targets inhibitory neurons releasing GABA or glycine. Purkinje and granular cells in the cerebellum express receptors for the neurotoxin and the purkinje cells release GABA as neurotransmitter, making them vulnerable to the toxin. This is in line with the observation of decreased Purkinje and granular cells in autopsies of autistic children [43].

The yeast Candida appears to play a role in autistic children as well. Strati et al. found significantly elevated abundances of the bacterial genera *Collinsella*, *Corynebacterium*, *Dorea* and *Lactobacillus* in the faeces of forty autistic children when compared to healthy children. Lowered levels of *Alistipes*, *Bilophila*, *Dialister*, *Parabacteroides* and *Veillonella* were also found in autistic children. Additionally, the authors found the yeast *Candida* to be present at increased rates in autistic children. Constipation could be associated with higher levels of *Escherichia*, *Shigella* and *Clostridium* cluster XVIII and lower levels of *Gemmiger* and *Ruminococcus* [87]. *Lactobacillus* spp stimulate the immune system to produce IL-22. IL-17 and IL-22 together inhibited the overgrowth of *Candida* spp, however in autistic population, the altered diversity of microbial community favoured the growth of *Candida* spp. Furthermore, *Candida* spp, once established in the gut, prevented recolonization of commensal microbes [87]. Several studies found notably higher abundance of *Candida* spp, especially *Candida albicans* in the faecal samples of autistic children compared to healthy counterparts [15,95]. Kantarcioglu et al. investigated the abundance of *Candida* spp in the gut of 415 autistic children (and 403 controls), which was elevated. Reportedly this yeast is associated with some autistic behaviour and 60 % of the healthy population is estimated to be asymptomatic carrier of *Candida* spp [95]. Normally, *Candida* cannot grow in the healthy microbial environment due to the competition for space and nutrients and suppression by commensal bacteria. However, in a dysbiotic environment as frequently observed in the autistic population, the yeast proliferates and produces ammonia and toxins, which were reported to increase autistic behaviour [95]. *Candida* spp also cause malabsorption of minerals and carbohydrates potentially playing a role in the ASD pathophysiology. 

A simulation by Weston et al. showed the interdependency between the anti-inflammatory genera *Bifidobacterium* and the pro-inflammatory *Clostridia* and *Desulfovibrio*. *Bifidobacterium* is inhibited by lysozyme and the growth of *Desulfovibrio*. To some extent, *Desulfovibrio* thrives on metabolites produced by *Bifidobacterium*. Growth of *Clostridia* is inhibited by lysozymes and by a higher abundance of *Bifidobacterium*. The authors claimed that the growth of *Clostridia* in the gut with low abundances of Bifidobacterium is a key risk for the development of ASD [96]. Another simulated study analysed the microbiome coding for enzymes involved in the metabolism of glutamate and found that these enzymes were underrepresented in autistic microbiome compared to healthy ones. Glutamate is a constituent of the important peptide glutathione, which is an antioxidant and therefore reduces oxidative stress in the cell. This amino acid is also an excitatory neurotransmitter. An imbalance in the CNS between excitation and inhibition has been postulated to contribute to ASD [97]. A general observed trend is that *Bifidobacterium* spp are scarcer represented in the guts of autistic children, whereas *Clostridia* spp are higher abundant [98].

Detecting altered microbial composition is especially important to understand how microbial metabolites can modulate gut and neuronal functions. Future studies are needed to establish a possible causation. Nevertheless, a significantly changed microbial composition was found in the GI tracts of autistic population pointing to a correlation between the microbiota and the occurrence of ASD. However, some of the mentioned results contradict each other. Conflicting results have been found for example for Oscillospira spp [40,89], Collinsella spp [40,87] and Parabacteroides spp [40,87]. Therefore, conclusive evidence is yet to be established by future studies.

### 5.2. Altered Concentrations of Metabolites and Their Functional Consequences

#### 5.2.1. Urinary Analysis

A study by Xiong et al. analysed urinary samples from sixty-two autistic and sixty-two non-autistic controls. They found significantly higher concentrations of 3-(3-hydroxyphenyl)- 3-hydroxypropionic acid, 3-hydroxyphenylacetic acid and 3-hydroxyhippuric acid in samples of autistic children (Table 1) [99]. These data are in line with the proposition that 3-(3-hydroxyphenyl)-3-hydroxypropionic acid induce autistic symptoms by decreasing catecholamine levels in the brain [38]. After oral administration of Vancomycin and supplementation of the probiotic *Bifidobacterium* to these children the mentioned metabolites significantly decreased [99]. Vancomycin is an antibiotic, which is poorly absorbed by the GI mucosa and has a great effect on gram-positive bacteria such as *Clostridia* spp. The decrease of 3-(3-hydroxyphenyl)-3-hydroxypropionic acid, 3-hydroxyphenylacetic acid and 3-hydroxyhippuric acid after treatment suggests that these metabolites are mostly produced by *Clostridia* spp and other overgrown populations in an altered phenylalanine metabolism. The treatment showed improved eye contact behaviour and less constipation in autistic children [99]. This finding was in line with the observation that children with altered phenylalanine metabolism, such as phenylketonuria, also often develop ASD [100].

Some Clostridiaceae spp are known to produce metabolites such as phenols, *p*-cresol and indoles, all of which may act as toxins on the human metabolism [40]. *P*-cresol, which was investigated as a possible biomarker for ASD, was elevated in the urinary [101,102] and faecal [40] samples of autistic children (Table 1). *P*-cresol has an inhibitory effect on the enzyme dopamine-beta-hydroxylase and thereby regulates the metabolism of the important neurotransmitter dopamine in the brain [100]. As *p*-cresol is only produced in the GI tract, its occurrence in the system correlated with increased permeability of the gut. In addition, *p*-cresol was shown to correlate with autistic behaviour and the severity of the disease [103].

Yap et al. compared urinary metabolite profiles of thirty-nine autistic children, twenty-eight non-autistic siblings and thirty-four healthy controls using NMR spectroscopy. Autistic children were found to excrete significantly more *N*-methyl-2-pyridone-5-carboxamide, *N*-methyl nicotinic acid, and *N*-methyl nicotinamide in their urine. Knowing that nicotinamide derives from tryptophan, this finding suggested ASD-associated alterations of the nicotinamide and the tryptophan metabolism [104]. Other significantly elevated metabolites in the urinary profile of ASD children were acetate, dimethylamine, *N*-acetyl glycoprotein fragments, succinate and taurine. Significantly lower levels of glutamate, hippurate and phenylacetylglutamine were present in the urinary profile of ASD children. Precursors of phenylacetylglutamine and hippurate are phenylacetic acid and benzoic acid, which are both produced by bacterial metabolism [104]. Hippurate is benzoic acid conjugated with glycine [105]. *N*-methyl-2-pyridone-5-carboxamide, *N*-methyl nicotinic acid, and *N*-methyl nicotinamide are end products of the nicotinamide pathway. It is worth mentioning that no major differences in the metabolic profile was observed comparing healthy siblings to autistic children or non-autistic controls. This indicated that siblings of autistic children exhibited a metabolic profile in between their autistic siblings and normal controls [104]. 

In their urinary metabolite profile of thirty autistic children Gevi et al. found major differences in the tryptophan and purine metabolism compared to thirty non-autistic children. Other significantly perturbed pathways were the vitamin B6 pathway and the phenylalanine- tyrosine-tryptophan biosynthesis, among others [101]. Additionally, increased levels of tryptophan degradation products were found in the urine suggesting increased production of tryptophan or increased abnormal degradation [106]. Several pathways utilize tryptophan in the human body (Figure 5). Indican or indolyl-lactate is produced by bacterial degradation. Through the human metabolism in the kynurenine pathway, kynurenic acid, xanthurenic acid or quinolinic acid are generated. Serotonin and melatonin are produced in another pathway from tryptophan (Figure 5). In the autistic metabolome kynurenic acid and melatonin are found in much lower concentration and xanthurenic acid and quinolinic acid are elevated compared to controls. The enzyme kynureninase uses vitamin B6 as cofactor. Therefore, B6 deficiency explains lower concentrations of kynurenic acid and elevated levels of xanthurenic acid and quinolinic acid in the kynurenine pathway [101]. Also, greater amounts of indolyl-3-acetic acid and indolyl-lactate were found in the urine [101] and indole and 3-methylindole in the faeces of autistic children [40]. Following species produce indoles: *Escherichia coli*, *Proteus vulgaris*, *Paracolobactrum coliform*, *Achromobacter liguefaciens*, *Bacteroides* spp [101] and *Clostridium* spp [40]. Indole is absorbed in the gut, thereafter it is oxidized and sulphated in the liver to indoxyl sulphate. Indoxyl sulphate may lead to an accumulation of several neurotransmitters in the brain, blocking the efflux transporter in the BBB [101]. A study revealed increased excretion of sulphate, sulphite and thiosulphate into the urine in ASD. They reasoned that this finding was due to dysfunctional sulphate transporters in the renal tube cells, hindering the reabsorption of these metabolites [104].

Other metabolites found in altered concentrations were TNF-α, free amino acids and quorum sensing molecules (Table 1). TNF-α was found in heightened levels in the faeces of autistic children, which correlated with the severity of GI symptoms [49]. Because increased TNF-α and IL-6 can also be a marker of inflammation in the brain [1], these data re-confirmed a higher level of inflammation in the brains of autistic children. Circulating free amino acid levels were elevated in patients with ASD. Glutamate especially was found in higher quantities in the faeces of autistic patients. Glutamate is a neurotransmitter and the excess of it and can lead to apoptosis of neurons [40]. Quorum sensing molecules such as PhrCACET1 are signal molecules exchanged between bacteria. Several *Clostridium* spp synthesize PhrCACET1, which can readily cross the BBB. *Clostridium* spp are highly abundant in autistic individuals, therefore, the effect of PhrCACET1 should be closer investigated [107].

Accumulatively, these data showed several altered metabolites in the urinary profile of autistic children, some even with correlation to autistic behaviour such as *p*-cresol. The metabolic pathways of tryptophan, purine, vitamin B6 and phenylalanine seem to be perturbed in ASD. Combination therapy with Vancomycin and *Bifidobacterium* led to amelioration of autistic symptoms such as improved eye contact and communication behaviour, assessed by the “Autism Behaviour Checklist” [99]. In addition, the above therapy resulted in normalized levels of the metabolites 3-(3-hydroxyphenyl)-3-hydroxypropionic acid, 3-hydroxyphenylacetic acid and 3-hydroxyhippuric acid in the urine of autistic individuals [99] providing the existence of a possible link between these metabolites and the pathology of ASD. These metabolites indicated an altered phenylalanine metabolism in ASD. Further studies are warranted to work out the true potential of these metabolites as possible biomarkers for ASD. 

#### 5.2.2. Blood Analysis

Elevated levels of several soluble factors including neurotransmitter were observed in the blood of autistic patients. Serotonin levels in the whole blood were elevated in 23 % of the autistic children (Table 2) [108]. In addition, co-occurring GI symptoms were observed in children with ASD and children with hyperserotonaemia. However, the relationship between hyperserotonaemia and constipation was not unequivocal in the study [108]. Serotonin serves as a major neurotransmitter in the CNS and the gut. In the GI tract, serotonin controls motility, pain perception and GI secretion. In the brain serotonin is responsible for regulating mood and cognition. In germ-free mice, higher levels of serotonin were found in the hippocampus and higher levels of tryptophan, the precursor of serotonin, in the blood [73]. Known producers of serotonin are species from *Candida*, *Streptococcus*, *Escherichia* and *Enterococcus* [72]. Another neurotransmitter found in elevated concentration in autistic children was the inhibitory neurotransmitter GABA [98]. Some *Lactobacillus* and *Bifidobacterium* spp are known producers of GABA [24,72,109]. Producers of another neurotransmitter, noradrenaline, are spp from *Bacillus*, *Enterococcus*, *Escherichia*, *Saccharomyces* and *Streptococcus* [24,72]. Neurotransmitters produced in the gut act upon gut epithelial cells to produce active molecules that may reach and influence the CNS after the secretion into the periphery or by activating afferent neurons.

Autistic children are also known to have significantly lower plasma concentrations of glutathione, homocysteine, methionine and *S*-adenosylmethionine (Table 2) [104]. *S*-adenosylmethionine acts as an important methyldonor in the body and homocysteine is a precursor of *S*-adenosylmethionine, suggesting perturbations in the sulphur metabolism. Methylation and glutathione are used to lower oxidative stress in the cell. The methylation of nicotinic acid to *N*-methyl-2-pyridone-5-carboxamide, *N*-methyl nicotinic acid, and *N*-methyl nicotinamide consumes even more methyl-donor leading to increased sensitivity to oxidative stress in cells of autistic children [104]. The excretion of heavy metals is difficult in ASD patients due to the fact that glutathione is less available and transmethylation/transsulphuration pathways are altered [86]. Acetaminophen (Paracetamol) is mal-tolerated by autistic individuals. Mal-toleration occurs also due to insufficient capability to sulphate and thus detoxify the drug and due to the abundance of higher levels of *p*-cresol in autistic children. This is in total agreement with the fact that non-autistic people with high *p*-cresol blood levels are also known to be intolerant to Paracetamol [104]. The above data indicates an alteration in the sulphur metabolism and a reduced reduction capability in the cells of autistic people owing to the decreased levels S-adenosylmethionine and its precursor homocysteine.

Examining metabolites in serum and in red blood cells in eleven Canadian autistic children revealed significantly lower concentrations of docosahexaenoic acid (DHA), eicosapentaenoic acid (EPA), arachidonic acid (ARA) and lower ratio of omega-3 to omega-6 fatty acids in red blood cells compared to fifteen controls (Table 2). Also, significantly lower concentrations of DHA, ARA and linoleic acid were found in the serum [110]. Decreased fatty acids levels in patients might be a direct result of the eating pattern observed in autistic individuals and not a cause leading to autism. As constituents of neuronal membranes, both DHA and ARA play a role in the maturation of neuronal networks. DHA is also important for the myelination of axons [111]. Both fatty acids are also modulator of gene expression in the cell, which could also be a contributing epigenetic factor for ASD [110]. In conclusion, the picky eating habit observed in autistic children might be a contributing factor to ASD. 

Another metabolite with altered concentration in ASD patients was thiamine-pyrophosphate (TPP). The concentration in the plasma was significantly decreased compared to controls. However, precursor concentrations (thiamine and thiamine-monophosphate) were similar in both groups (Table 2) [112]. Lower concentration of TPP could result from an impaired excretion from producing cells into the blood or an impaired uptake from the gut lumen. Indeed, over 40% of the gut microbiota are known to produce TPP. TPP is a cofactor to many enzymes in the mitochondria, most importantly transketolase in the oxidative pentose-phosphate pathway [112]. A decreased function of transketolase leads to an accumulation of the substrate for the oxidative pathway, which in turn inhibits the reductive pathway of pentose-phosphate-pathway resulting in lower concentration of NADPH. Lower abundance of NADPH leads to an increased oxidative stress in the cell [112]. Thus, decreased levels of TPP lead to a lessened reduction capacity, as they are a cofactor of transketolase inhibiting the production of NADPH. In summary, the reduced anti-oxidative capacity may lead to a cellular damage and a reduction of mitochondria’s capacity to produce energy in ASD individuals. Increased amounts of LPS were also found in the blood of autistic individuals (Table 2) [56,113]. LPS is produced by gram-negative bacteria and exerts pro-inflammatory activity. Leakage through the BBB can cause inflammation in the brain through activating microglia. Heightened LPS levels in autistic individuals correlated with elevated IL-6 levels, another pro-inflammatory cytokine. An inflammation further compromises the integrity of the BBB, facilitating the accumulation of heavy metals in the brain tissue, including the cerebellum [20,56], thus worsening the symptoms. 

#### 5.2.3. Short Chain Fatty Acids

SCFAs, mainly butyrate, propionate and acetate [114], are produced as fermentation products by gut bacteria [115,116]. Lower concentrations of the total amount of SCFAs are found in autistic children [40,98], providing the evidence for a reduced fermentation capacity of the microbiota. Nevertheless, propionate and acetate are found in higher levels in autistic children suggesting that butyrate production is dramatically reduced [40]. SCFAs cross the BBB and influence early brain development by modulating production of the neurotransmitter serotonin and dopamine [40]. These acids bind to two free fatty acid receptors (FFA2 and FFA3) found in several areas of the brain [6,72,117]. The resorption of water and electrolytes in the colon is also to some degree regulated by SCFAs [40]. They also modulate the cytokine production by T-cells [62]. Positively correlated with total amount of SCFA were *Faecalibacterium*, *Ruminococcus* and *Bifidobacterium* genera, while the amount of *Bacteroides* spp was correlated with propionic acid in autistic individuals [40]. *Bacteroides* spp, *Porphyromonadaceae* and *Parabacteroides* were positively correlated with the amount of propionate and butyrate in autistic individuals. Alcaligenaceae is also positively correlated with propionate, whereas the Bacteroidaceae is negatively correlated with acetate and total amount of SCFA in autistic children [118]. 

Intraventricular injection of propionate into rat pups caused behavioural and physiological changes as observed in ASD [98]. An accumulation of propionic acid can lead to prolonged neurodevelopment and seizures [16]. *Clostridia* spp are known to produce exotoxins and propionate and thus create an inflammatory status, which might worsen autistic symptoms [87]. As mentioned before, presence of an oxidative milieu in the cell, as seen in autistic individuals, leads among others to malfunctioning of mitochondrial enzymes and the production of reactive nitrogen species [119]. The reactive nitrogen species interact with propionic acid and produce 3-nitropropionic acid. This acid acts as a mitochondrial neurotoxin, which irreversibly inhibits the function of the enzyme succinate dehydrogenase, which is implicated in the synthesis of NADH. Propionic acid is also converted to propionyl-CoA and enters the citric acid cycle at the enzyme succinyl-CoA-dehydrogenase. Two enzymes which are important to produce NADH are surpassed by 3-nitropropionic acid [119]. NADH is a substrate for the electron transport chain and consequently for ATP production. This explains the lower activity of ETC enzyme complex I due to reduced substrate concentration [119]. Propionate is involved in up-regulating the cyclic-adenosine-monophosphate response element binding protein (CREB). CREB regulates synaptic plasticity, memory formation and the reward system [120]. A study by Frye et al. compared mitochondrial function of lymphoblastoid cells from autistic subjects and controls after incubating them in propionic acid and introduction to reactive oxygen species (ROS). Propionic acid metabolised by healthy lymphoblastoid cells exerted beneficial effects on the cells such as increased respiration. Cells from autistic individuals consumed propionic acid much faster as a fuel, suggesting a compensatory measurement for energy production. The consequence of the stronger reaction to ROS after propionic acid incubation by lymphoblastoid cells of autistic subjects was, among others, an increased proton leakage from the mitochondria to the inter-membrane space and hence a reduced respiratory capacity [119].

Butyrate is used as an energy source in colonocytes [62]. It is involved in regulating inflammatory and oxidative state of the mucosal cells, visceral sensitivity and motility and reinforces the mucosal barrier in the intestines. It prevents carcinogenesis and inflammation in the gut via its anti-inflammatory properties [114]. Butyrate modulates the biosynthesis of catecholamines and neurotransmitters in the CNS and autonomic nervous system [106]. Butyrate is said to be the most important SCFA in humans as it exhibits neuroprotective features. It promotes memory formation and neuronal plasticity through epigenetic modulation [121]. In some neurodegenerative models butyrate was found to ameliorate symptoms [121]. Even if not conclusive, the above data suggest an involvement of increased acetate and propionate and decreased butyrate in the pathophysiology of ASD.

### 5.3. Other Etiologies for ASD

Over one hundred genes are associated with the pathology of ASD and up to several hundred genes are implicated in elevation of the susceptibility to ASD [38]. Most of the affected genes are implicated in synapse plasticity [122] and brain development in utero as well as during infancy [27]. Most of the differing gene expression lead to an altered structure of the CNS or the enteric nervous system [38]. ASD has been hypothesised to be linked to abnormalities of the neural crest development during pregnancy [123]. Inoue et al. found in peripheral blood mononuclear cells of autistic children, 1056 up-regulated and 517 down-regulated genes, respectively [124]. Comorbidities such as fragile X syndrome, tuberous sclerosis or Rett disorder have also been observed in ASD patients (Figure 2) [27]. Six Japanese autistic children were compared to six healthy controls regarding microbial composition by collecting faeces and blood. The genus *Faecalibacterium* was higher and *Blautia* lower represented in the faeces of autistic children [124]. Moreover, there were several up- and down regulated genes in autistic children compared to healthy ones. Interestingly some genes, which are involved in IFN-gamma mediated pathways correlated with the differing abundances of *Faecalibacterium* and *Blautia* [124]. These data are in accord with the finding that IFN-gamma is elevated in autistic children, which suggests a chronic inflammation in these children.

Another aetiology for ASD-like behaviour in mice was the exposure to bisphenol A *in utero*. Bisphenol A changed the DNA methylation and thus, the expression level of the gene for BDNF [125]. BDNF is important for survival of neurons, growth and differentiation of newly formed neurons as well as new formations of synapses [72]. 

The data above indicates that several genes, which are involved in CNS development and inflammation pathways, are altered in autistic individuals providing several targets for a better understanding of the mechanisms involved in the disease and possible therapeutic targets. 

## 6. Possible Therapeutic Measures Acting on the Microbiota

### 6.1. Prebiotics

Dietary prebiotics were defined as “a selectively fermented ingredient that results in specific changes in the composition and/or activity of the gastrointestinal microbiota, thus conferring benefit(s) upon host health” [77,126].

Grimaldi et al. investigated the role of the prebiotic galactooligosaccharide (GOS) in a synthetic colon model comparing diluted faecal samples from autistic children and controls regarding changes in bacterial composition and metabolic profile. The administration of the prebiotic led to significant increases in *Bifidobacteria* spp in both autistic and control models [98]. Slightly significant increases of the *Clostridium* cluster XI and decreases of Veillonellaeceae were observed only in the ASD group. Significant decreases occurred in *Sutterella* spp, *Bacteroides*, Clostridial cluster IX, *Escherichia coli*, Veillonellaceae, *Ruminococcus* spp, *Clostridium leptum* and *Faecalibacterium prausnitzii* in autistic children. Growth of *Bifidobacteria* could be stimulated by the prebiotic [98]. GOS also affected SCFA production, increasing butyrate and decreasing propionate. Thus, it is sensible to postulate that the effects of the GOS supplementation in these children were, at least partially, through the production of acetate and butyrate [98]. 

Prebiotics are not widely researched as a treatment option for autistic individuals. Nonetheless, existing data point to the fact that prebiotic treatment can cause microbial compositional alterations in autistic children that could result in amelioration of autistic symptoms. As mentioned before, methodologies to deliver therapeutics, such as specific SCFA or prebiotics, to the GI tract could be envisaged to directly target the colon to reach the microbiota without the interference of the digestive system.

### 6.2. Probiotics

Probiotics contain living microorganisms and are administered to promote health by stimulating immunity, strengthening the intestinal barrier, increasing the expression of mucine, reducing the overgrowth of pathogens and producing vitamins and antioxidants [1,127]. The most commonly used probiotics are Bifidobacterium spp and Lactobacillus spp [56]. 

The administration of the probiotic cocktail “Children Dophilus”, containing strains from *Lactobacillus*, *Bifidobacterium* and *Streptococcus* [49], three times a day for four months shifted the Bacteroidetes/Firmicutes ratio towards more Bacteroidetes (Table 3). The abundances of the genera Desulfovibrio and Bifidobacterium also increased in autistic children. The administration of the probiotic mixture of *Lactobacillus rhamnosus*, *Bifidobacterium infantis*, *Bifidobacterium longus*, *Lactobacillus helveticus*, *Lactobacillus reuteri* and *Lactobacillus paracasei* to autistic children improved comorbid GI problems [106]. *Lactobacillus reuteri* on its own could reverse intestinal inflammation caused by LPS [46]. Furthermore, the bacterium *Lactobacillus reuteri* influenced the posterior pituitary gland to produce more oxytocin. Oxytocin is a hypothalamic hormone positively influencing social behaviour [24]. It remains to be determined whether increased levels of oxytocin in autistic children could ameliorate their behaviour in social interactions in a well-designed experimental setting.

The administration of *Bifidobacterium* spp to mice for 8 weeks resulted in higher concentration of ARA and DHA in their brains [117]. This altered fatty acids composition in the brain exerted a beneficial effect on neurogenesis, neurotransmission and preventing oxidative stress. Learning and memory were also positively affected by these fatty acids [117]. The administration of *Lactobacillus rhamnosus* in mice, on the other hand, reduced corticosterone levels and thereby positively influenced anxiety and depression. Another observed effect was altered levels of GABA and its receptors throughout the brain. These effects could only be achieved with an intact vagal nerve, a fact that reconfirms the role of the vagal nerve as a major route of communication between the brain and the gut microbiota [117].

Hsiao et al. treated gut permeability and altered microbial composition by oral administration of the commensal microbe *Bacteroides fragilis* in MIA mice showing improvements of autistic features as well as amelioration of deficits in anxiety-like behaviour. Interestingly the probiotic also corrected altered expression of the tight junction proteins CLDN-8 and CLDN-15 in the colon. Heightened levels of IL-6 found in the colon of MIA mice were normalised, which led to improved gut barrier function. A metabolite, which was found in significantly higher levels in the serum of MIA mice is 4-ethylphenylsulfate, which is chemically similar to *p*-cresol and is produced by *Clostridia* spp. The authors convincingly showed that the oral administration of *Bacteroides fragilis* corrected serum levels of 4-ethylphenylsulfate in MIA mice [5]. 

Probiotics are known to alter the microbial composition in the gut of autistic children. The experimental evidence for the positive behavioural changes observed in MIA mice after administration of *Bacteroides fragilis* raise the interesting question whether the same result also applies to humans [128].

### 6.3. Faecal Microbiota Transplantation

In contrast to probiotic treatment, where just some bacterial strains are supplemented, faecal microbiota transplantation (FMT) ensures the transfer of several hundred bacterial strains. A healthy individual donates a stool sample [79], which is normally used freshly within eight hours or within eight weeks when directly frozen after donation [129]. FMT could potentially be problematic, as donors could transfer opportunistic pathogens or infections to recipients. A thorough screening of donors before donation minimises the risk but in the early stages of infections donors could be asymptomatic and unknowingly transfer infections [130]. Despite these uncertainties, FMT is an established and efficient therapy in recurrent *Clostridium difficile* infection [77,79,131,132].

Kang et al. studied the impact on the microbial composition and the course of GI and autistic symptoms in eighteen autistic children after “Microbiota Transfer Therapy” [79]. The transfer followed over a period of 7–8 weeks after oral administration of Vancomycin and bowel cleansing. The results were impressing, because GI symptoms such as constipation, diarrhoea or abdominal pain decreased by 80%. The improvement lasted at least for eight weeks. This study showed clearly an improvement of not only GI but also seventeen ASD behavioural symptoms assessed by the “Parental Global Impressions-III” after FMT [79]. The main consequence of faecal transplantation was an increased bacterial diversity and altered abundances of some bacteria such as the genera *Bifidobacterium*, *Prevotella* and *Desulfovibrio*. This suggests that GI and behavioural symptoms in autistic children are inter-connected and trace back to lesser microbial diversity probably through modulating metabolites [79].

Therefore, FMT is an exciting new treatment option, which could be considered for ASD patients with dysbiosis.

### 6.4. Other Treatment Strategies

Amelioration of behavioural symptoms could also be reached through oral treatment with Vancomycin, which is an antibiotic specific to bacteria such as Clostridiaceae [40]. This finding suggests a connection between Clostridiaceae and autistic behaviour. However, the amelioration of the symptoms did not last longer than the treatment duration [40]. This was attributed to the spore forming capacity of these bacteria as bacterial spores can survive antibiotic treatment and proliferate afterwards [40]. Also, a restricted gluten and/or casein-free diet seemed to positively affect the social and cognitive problems in children with ASD [40]. One hypothesis is that these food components are metabolised to opioid-like substances and act in the CNS [106,133]. However, the evidence is rather weak, as opioid-like substances were not found in autistic individuals. Therefore, the therapeutic value of omitting gluten and casein free diet is limited and should only be considered if there are allergic reactions to the abovementioned food components [86,106].

Interestingly, higher rates of ASD have been reported in people with dark skin and in regions with limited solar irradiation. Significantly lower levels of 25-hydroxy-Vitamin D in serum were found in autistic children in comparison to controls. In a study vitamin D supplementation was given to rat pups to treat ASD induced by propionic acid. The authors discovered that vitamin D has neuroprotective features and less DNA breaks were detected in the brains of rats treated with vitamin D compared to untreated propionic acid induced rats. These results, however, could only be obtained, when the pups were treated with vitamin D before inducing autism. In conclusion, the study suggested that vitamin D exhibits rather a preventive than a therapeutic effect [134]. The administration of vitamin B1 to autistic children was also suggested to ameliorate symptoms due to vitamin B1 producing bacteria Prevotella being found in lower abundances in autistic children [20]. 

One publication even suggested the introduction of helminths, which are parasitic worms, into the gut ecosystem to achieve a greater diversity. One argument was that before daily hygienic measures were introduced in our society, helminths were a normal part of the gut microbiota and the prevalence for ASD was smaller [123]. This hypothesis was experimentally supported by the data that the exposure of the immune activated rats to helminths resulted in less anxious behaviour [123], which is considered to be a symptom of ASD in rodents [102]. Obviously, these treatment options were poorly documented and are not recommended as a treatment option for ASD patients before thorough investigations. 

## 7. Discussion and Conclusions

It remains yet to be unequivocally determined whether dysbiosis is a factor causing ASD or if the disease is causing the microbial alterations. The mutual correlation between ASD and alterations in the microbiota, however, has been undoubtedly confirmed by many animal and human studies [124]. To further narrow down the mechanisms of disease, studies with bigger sample sizes are required, since the works summarized in this review showed varying degrees of statistical significance and some results are based on small sample sizes. Our review also showed big variations of methodologies and outcome measures calling for standardisation of the treatment regimens to achieve higher statistical significance and a better comparison between the study results.

This review article demonstrated significant changes in microbial composition of autistic population. Often observed changes in ASD children were a generally decreased bacterial diversity compared to control populations and the significantly decreased Bacteroidetes to Firmicutes ratio [49,53,76,87]. Elevated abundances of *Clostridium* spp in autistic individuals correlated with the severity of autistic behaviour [15]. An evolutionary explanation as to why the microbiota modulate our behaviour is, the more social the host is, the more the microbes can spread through society [135].

The discovery and understanding the connection between the microbiota and ASD is especially important to find new treatment options for this disease. Treatment of autistic behaviour based on microbial compositional alterations such as probiotics and faecal microbiota transfer showed promising results in GF-mice and autistic individuals [5,46,49,79,106,117]. The administration of probiotics resulted in improvement of comorbid GI problems [106] or stimulation of the production of oxytocin, positively influencing social behaviour [24,136]. Especially, faecal microbiota transfers significantly ameliorated ASD behavioural symptoms and GI problems in autistic children [79]. For widespread use of these treatments more studies must be conducted to finetune treatment, mode of delivery, safety, its duration and doses. 

We suggest that the reduced mRNA expressions of disaccharidases and hexose transporters in the gut epithelium may be responsible for some of the alteration in microbial composition of ASD patients [53]. More mono- and disaccharides entering the colon favour some bacterial species over others and consequently lead to an altered metabolome in the gut affecting the host. Such alterations may result in different expression levels of tight junction proteins [55] leading to comorbidities such as increased intestinal permeability [15] and a reduced integrity of the BBB in ASD patients. As a result of these changes, bacterial metabolites can enter and influence the CNS.

Significantly lower plasma concentration of S-adenosylmethionine, glutathione [104] and thiamine-pyrophosphate [112] in ASD provide the evidence for a perturbation in transmethylation processes in the body and diminished reduction capability of the cell. Increased oxidative stress in the cells leads to malfunctioning of mitochondrial enzymes and production of reactive nitrogen species, which can interact with propionic acid and produce a mitochondrial neurotoxin worsening the reduction capability [119]. Lower concentrations of the total amount of SCFAs are found in ASD patients [40,98] with higher levels of propionate and indicating that butyrate production is dramatically reduced [40]. The altering concentrations of the bacterial metabolites butyrate and propionate are likely to play a role in the pathophysiology of autism. A new possible family of substances with link to ASD is suggested in our review including 3-(3-hydroxyphenyl)-3-hydroxypropionic acid, 3-hydroxyphenylacetic acid and 3-hydroxyhippuric acid. These metabolites were increased in the urinary profile of ASD patients linking ASD to an abnormal phenylalanine metabolism. As a consequence of treating ASD patients with an antibiotic and a probiotic, these metabolites decreased in the urine of autistic patients. Additionally, autistic behaviour was ameliorated in the patients due to this treatment [38]. The evidence for the predictive value of the suggested biomarkers is based on one well conducted study only. Therefore, the effectiveness of these molecules for prediction of ASD need to be validated in independent human studies, to reconfirm the finding that these metabolites are part of an autistic-behaviour-causing metabolome. 

In conclusion, the correlation between changes in distinct bacterial populations and several bacterial metabolites, and the behavioural changes related to ASD warrant further investigations into the microbiota-gut-brain-axis aiming at in-depth examination of mechanisms leading to the pathology of autism. We presented here several epidemiological, experimental human and in vivo studies for the involvement of different microbes in the ASD pathology such as *Clostridia* spp and highlighted the evidence for several metabolites to be considered as possible biomarkers for ASD. Additional independent research following standardised protocols, and enrolling well-diagnosed bigger cohorts are urgently needed to definitely discover the interdependencies, the sequence of events leading to ASD, and to suggest unequivocal biomarkers and effective therapeutic strategies.

## Figures and Tables

**Figure 1 ijms-20-02115-f001:**
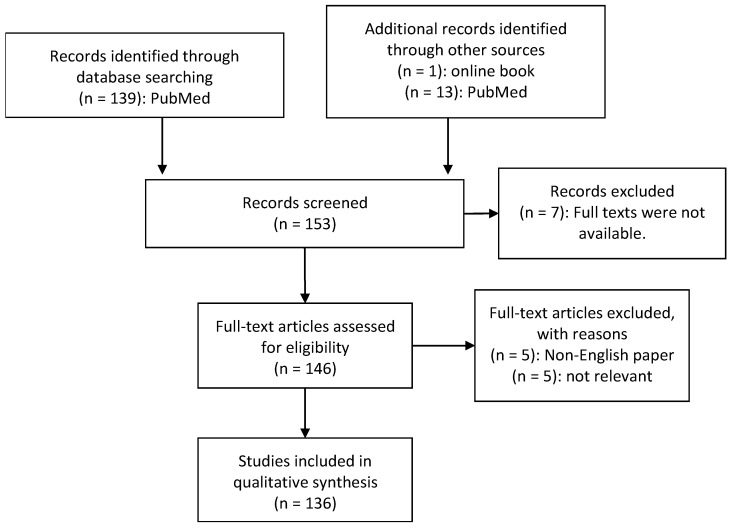
Methodical approach of this systematic review due to PRISMA criteria [39].

**Figure 2 ijms-20-02115-f002:**
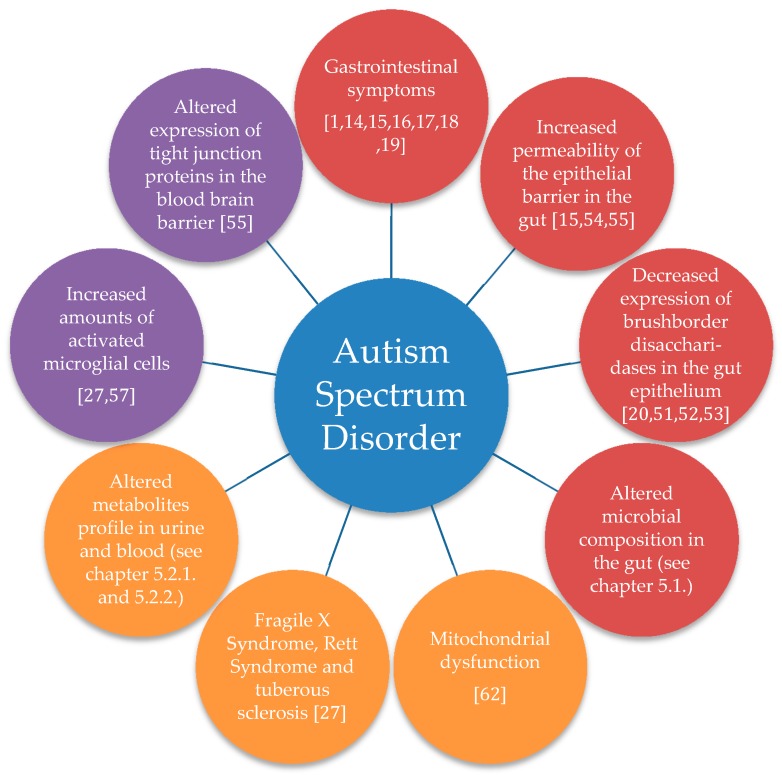
Co-occurring pathologies found in Autism Spectrum disorder (ASD). Red shows gut-related comorbidities found in ASD, purple shows brain-related comorbidities and orange other comorbidities. Chapter 3 gives more information to the mentioned comorbidities.

**Figure 3 ijms-20-02115-f003:**
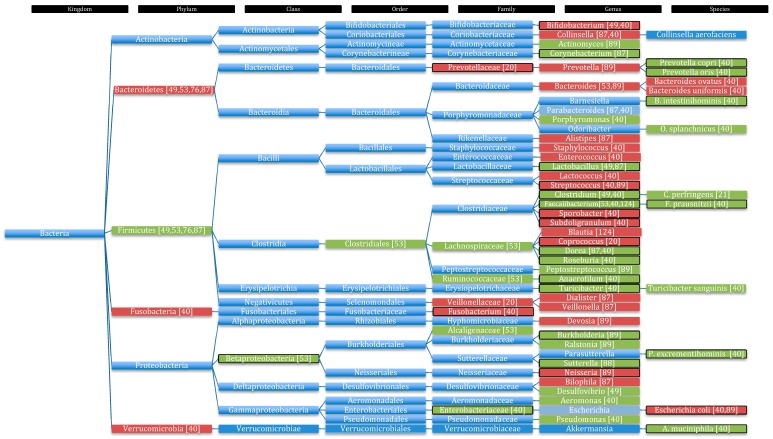
Taxonomic classification of dysbiotic bacteria found in autistic individuals. All bacterial taxa mentioned within this chapter are found in this figure. Green indicates elevated taxa in the gut microbiota of ASD individuals. Red indicates decreased taxa in the gut microbiota ASD individuals. Additionally, black surrounded boxes show significant results in at least one study. Significance levels are taken from the original literature.

**Figure 4 ijms-20-02115-f004:**
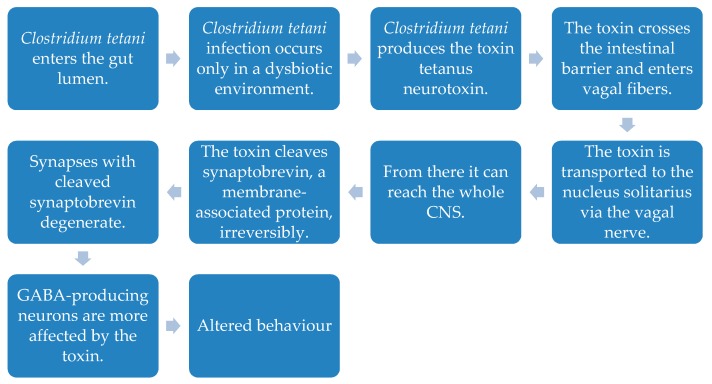
Process of *Clostridium tetani* infection in the gut. Occurrence of the bacterium *Clostridium tetani* in the GI tract of autistic population could lead to altered behaviour. [43].

**Figure 5 ijms-20-02115-f005:**
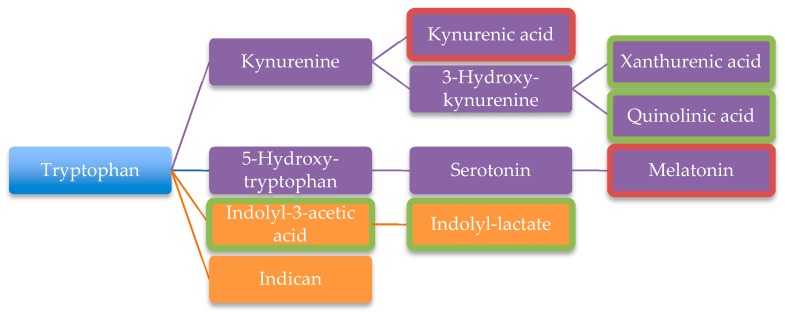
Metabolic pathways of tryptophan. The purple pathways show human metabolism of tryptophan and the orange pathways are observed in bacterial degradation. Green surrounded boxes show elevated metabolites and red surrounded boxes show decreased metabolites in the autistic metabolome [101].

**Table 1 ijms-20-02115-t001:** Elevated and decreased metabolites in the urinary profiles of ASD children. Significant results are written in bold.

Elevated Urinary Metabolites in ASD Patients		Decreased Urinary Metabolites in ASD Patients	
***N*-methyl-2-pyridone-5-carboxamide**	[104]	**Glutamate**	[104]
***N*-methyl nicotinic acid**	[104]	**Hippurate**	[104]
***N*-methyl nicotinamide**	[104]	**Phenylacetylglutamine**	[104]
***N*-acetyl glycoprotein fragments**	[105]		
**Succinate**	[105]	Melatonin	[101]
**Acetate**	[105]	Kynurenic acid	[101]
**Taurine**	[105]		
**Dimethylamine**	[105]		
**3-(3-hydroxyphenyl)-3-hydroxypropionic acid**	[99]		
**3-hydroxyphenylacetic acid**	[99]		
**3-hydroxyhippuric acid**	[99]		
Tryptophan degradation products	[106]		
Xanthurenic acid	[101]		
Quinolinic acid	[101]		
Indolyl-3-acetic	[101]		
Indolyl-lactate	[101]		
*P*-cresol	[101,102]		
TNF-α	[49]		
Free amino acids	[40]		
Sulphate, sulphite, thiosulphate	[104]		

**Table 2 ijms-20-02115-t002:** Summary of statistically significantly elevated and decreased metabolites in the blood of autistic patients. Major bacteria responsible for the elevated metabolites are given in parentheses.

Elevated Metabolites in the Blood of ASD Patients		Decreased Metabolites in the Blood of ASD Patients	
**Serotonin (Candida, Streptococcus, Escherichia and Enterococcus spp)**	[108]	**Methionine**	[104]
**GABA (Lactobacillus and Bifidobacterium spp)**	[98]	**S-adenosylmethionine**	[104]
***p*-cresol (Clostridia spp)**	[104]	**Homocysteine**	[104]
**Lipopolysaccharides (gram negative bacteria)**	[56,113]	**Glutathione**	[104]
		**Docosahexaenoic acid**	[110]
		**Eicosapentaenoic acid**	[110]
		**Arachidonic acid**	[110]
		**Thiamine-pyrophosphate**	[112]

**Table 3 ijms-20-02115-t003:** Non-exhaustive list of reports studying the effects of probiotics supplementation in ASD.

Probiotic Used	Effects	Studies
**Children Dophilus (Bifidobacterium, Lactobacillus, Streptococcus spp)**	Amelioration of Firmicutes to Bacteroidetes ratio	Humans [49]
**Lactobacillus rhamnosus, Bifidobacterium infantis, Bifidobacterium longus, Lactobacillus helveticus, Lactobacillus reuteri and Lactobacillus paracasei**	Improvement of GI problems	Humans [106]
**Lactobacillus reuteri**	- Reversing inflammation caused through lipopolysaccharides - Stimulating the production of Oxytocin	Rats [46]
**Bifidobacterium spp**	Elevation of arachidonic and docosahexaenoic acid concentration in the brain	Mice [117]
**Lactobacillus rhamnosus**	- Reduction of corticosterone levels - Alterations of GABA levels	Mice [117]
**Bacteroides fragilis**	- Improvement of autistic features - Correction of tight junction proteins expression in the colon	Mice [5]

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
