# Peer review of "The Possible Role of the Microbiota-Gut-Brain-Axis in Autism Spectrum Disorder"

_ijms, 2019, doi:10.3390/ijms20092115_

Round 1
Reviewer 1 Report
To my mind, this new version of the manuscript has significantly been improved by the authors:
The authors have included a "material and methods section" with a clear flowchart (inclusion and exclusion criteria) as any systematic review needs to present.
I am still missing the specific "question" the authors wants to answer based on their objective search and posterior analysis. This may be the reason why their final selection is too wide with 134 papers.
Still, after the material and methods section, the paper is presented as a "narrative review" and in this format reads as a very nice piece with interesting blocks of information and could be accepted for publication (even if the present manuscript may not follow the "orthodoxal" structure of a systematic analysis). In this case, the authors should coment and cite at the introduction section previously published systematic reviews already linking gut bacteria and ASD (see below). This would allow the authors to highlight (and readers to differentiate) the new additions of present review paper when compared with previous similar literature.
Clostridium Bacteria and Autism Spectrum Conditions: A Systematic Review and Hypothetical Contribution of Environmental Glyphosate Levels.
Argou-Cardozo I, Zeidán-Chuliá F.
Med Sci (Basel). 2018 Apr 4;6(2). pii: E29. doi: 10.3390/medsci6020029. Review.
Clostridium Bacteria and its Impact in Autism Research: Thinking “Outside the Box” of Neuroscience.
Zeidán-Chuliá, F.; Fonseca Moreira, J.C. .
Commun. Disord. Deaf Stud. Hearing Aids 2013, 1, 101.
Author Response
Dear Editor,
We would like to thank the Referee for her/his positive evaluations of our manuscript as well as for the valuable comments that helped us improve the paper.
We have now included and briefly discussed the two additional publications suggested by this reviewer on lines 46-47 and 389-390 and adapted Figure 1 and literature citations in the text accordantly.
We trust that the revised version of our paper is now suitable for publication and we look forward to hearing from you at your earliest convenience.
Yours sincerely,
MH Mohajeri, PhD

Reviewer 2 Report
The authors responded to the reviewers's comments.
Author Response
Thank you very much for your positive Evaluation of our manuscript.
Yours sincerely,
MH Mohajeri, PhD